# Individual Influence of Trochlear Dysplasia on Patellofemoral Kinematics after Isolated MPFL Reconstruction

**DOI:** 10.3390/jpm12122049

**Published:** 2022-12-12

**Authors:** Andreas Frodl, Thomas Lange, Markus Siegel, Hans Meine, Elham Taghizadeh, Hagen Schmal, Kaywan Izadpanah

**Affiliations:** 1Department of Orthopedics and Traumatology, Freiburg University Hospital, 79106 Freiburg, Germany; 2Department of Radiology, Medical Physics, Freiburg University Hospital, 79106 Freiburg, Germany; 3Fraunhofer Institute for Digital Medicine, 28359 Freiburg, Germany; 4Department of Orthopedic Surgery, University Hospital Odense, Sdr. Boulevard 29, 5000 Odense, Denmark

**Keywords:** cartilage contact area, patella dynamics, trochlea dysplasia

## Abstract

Introduction: The influence of the MPFL graft in cases of patella instability with dysplastic trochlea is a controversial topic. The effect of the MPFL reconstruction as single therapy is under investigation, especially with severely dysplastic trochlea (Dejour types C and D). The purpose of this study was to evaluate the impact of trochlear dysplasia on patellar kinematics in patients suffering from low flexion patellar instability under weight-bearing conditions after isolated MPFL reconstruction. Material and Methods: Thirteen patients were included in this study, among them were eight patients with mild dysplasia (Dejour type A and B) and five patients with severe dysplasia (Dejour type C and D). By performing a knee MRI with in situ loading, patella kinematics and the patellofemoral cartilage contact area could be measured under the activation of the quadriceps musculature in knee flexion angles of 0°, 15° and 30°. To mitigate MRI motion artefacts, prospective motion correction based on optical tracking was applied. Bone and cartilage segmentation were performed semi-automatically for further data analysis. Cartilage contact area (CCA) and patella tilt were the main outcome measures for this study. Pre- and post-surgery measures were compared for each group. Results: Data visualized a trending lower patella tilt after MPFL graft installation in both groups and flexion angles of the knee. There were no significant changes in patella tilt at 0° (unloaded pre-surgery: 22.6 ± 15.2; post-surgery: 17.7 ± 14.3; *p* = 0.110) and unloaded 15° flexion (pre-surgery: 18.9 ± 12.7; post-surgery: 12.2 ± 13.0; *p* = 0.052) of the knee in patients with mild dysplasia, whereas in patients with severe dysplasia of the trochlea the results happened not to be significant in the same angles with loading of 5 kg (0° flexion pre-surgery: 34.4 ± 12.1; post-surgery: 31.2 ± 16.1; *p* = 0.5; 15° flexion pre-surgery: 33.3 ± 6.1; post-surgery: 23.4 ± 8.6; *p* = 0.068). CCA increased in every flexion angle and group, but significant increase was seen only between 0°–15° (unloaded and loaded) in mild dysplasia of the trochlea, where significant increase in Dejour type C and D group was seen with unloaded full extension of the knee (0° flexion) and 30° flexion (unloaded and loaded). Conclusion: This study proves a significant effect of the MPFL graft to cartilage contact area, as well as an improvement of the patella tilt in patients with mild dysplasia of the trochlea. Thus, the MPFL can be used as a single treatment for patient with Dejour type A and B dysplasia. However, in patients with severe dysplasia the MPFL graft alone does not significantly increase CCA.

## 1. Introduction

Patellar instability is a frequent cause of early degenerative articular cartilage damage and developing patellofemoral osteoarthritis. Among others, trochlear dysplasia, patella alta and genu valgum are reported to predispose for patellar instability or recurrent dislocations [1]. However, patellar dislocations and instability are always accompanied by damage to the medial capsuloligamentous restrain tissue [2], consisting of mainly the medial patellofemoral ligament (MPFL), the weaker medial patellotibial ligament (MPTL), the medial patellomeniscal ligament (MPML), and the medial quadriceps tendon femoral ligament (MQTFL) [3,4]. Current studies report the MPFL as the strongest restraint against lateral translation of the patella and MPFL-plasty as a successful treatment option for patients with low flexion patellar instability [5,6]. About 50–60% of the total patellar restraining forces is provided by the MPFL between angles of 0–30° flexion [7]. The patellar tracking mechanism changes from predominant soft tissue, especially ligamentary restraints, to increasingly osseous tracking due to the engagement of the patella in the trochlear groove at about 30° of knee flexion [8].

The effects of surgical stabilization on patellar kinematics and cartilage contact mechanisms are being investigated. However, the impact of trochlear dysplasia on cartilaginous contact areas and patella kinematics after MPFL reconstruction remain unclear [9]. In order to achieve optimal post-surgery results and prevent early joint degeneration due to persistent maltracking after single MPFL reconstruction surgery, the necessity of adjunctional measures, i.e., trochlea augmentation, has to be evaluated.

Although evaluating joint kinematics and cartilage surface areas is challenging, it is greatly facilitated by fluoroscopy and Magnetic Resonance Imaging (MRI) [10,11]. Cadaveric studies primarily enabled insights into cartilage deformation under axial loading [12]. The reproducibility and quantification of MRI-guided investigations in vivo is technically difficult. Knee flexion is required to load the patellofemoral joint axially. Weight bearing under knee flexion leads to joint shaking, and consequently to motion artefacts in MRI scans. Pulley or pneumatic loading devices have been used to enable reproducible and comparable loading among a cohort of subjects [13,14,15,16]. In anticipation of motion artefacts, prospective motion correction (PMC) systems with optical tracking have successfully minimized artefacts during loaded MRI scans [16].

The purpose of this study was to evaluate the impact of trochlear dysplasia on patellar kinematics in patients suffering from low flexion patellar instability under weight-bearing conditions after isolated MPFL reconstruction. To test our hypothesis—that in patients with severe trochlea dysplasia, reconstruction of the MPFL does not sufficiently increase cartilage contact area (CCA) and patella tracking as “stand alone” treatment—we conducted pre- and post-surgery MRI scans of each patient to observe the intra-patient changes after MPFL augmentation.

## 2. Materials and Methods

Thirteen patients with a dysplastic trochlea and chronic patellar luxation were included in this study. Only patients with chronic patella instability, dysplasia of the trochlea, no previous surgery on the affected knee and aged between 18–40 years were included. Exclusion criteria included a history of prior patellofemoral surgery, metallic material from previous knee surgery, pregnancy, retropatellar osteoarthritis and claustrophobia. Measurements were taken pre- and post-surgery after implementation of an MPFL graft. Trochlear configurations and patellar positions were determined by measuring the patellotrochlear index, lateral trochlear inclination angle, mean osseous sulcus angle and then classified according to the Dejour classification for dysplastic trochlea in all subjects [17,18,19,20]. An overview of the patient selection process is shown in Figure 1.

The MRI measurements were performed with a Magnetom Trio 3T MRI system (Siemens Healthineers, Erlangen, Germany), using an 8-channel multipurpose coil (NORAS MRI products, Germany) for signal reception, which was attached to the thigh with a hook-and-loop fastener. The experiments were conducted with an MRI-compatible pneumatic loading device to enable accurate load adjustment in the 0–500 N range (Figure 1). A 3D turbo-spin echo (TSE) protocol with GRAPPA parallel imaging acceleration by a factor of 2 and an isotropic resolution of 0.5 mm was applied for the MRI scans. Further scan parameters were TR = 1.8 s, TE = 59 ms, receiver bandwidth = 504 Hz/Px, and scan duration = 6:20 min. For prospective motion correction, we used a moiré phase tracking (MPT) system with one in-bore camera and a single tracking marker [21]. Translational and rotational motion was optically tracked with a frame rate of 80 frames/s, enabling real-time updates of the MRI measurement volume before every excitation pulse.

The patient was tied to the scanner bed with a weight-lifting belt, and the leg to be examined was placed in the sliding carriage of the loading device (Figure 2). Measurements were performed with knee flexion angles of 0°, 15° and 30°. For accurate and reproducible angle adjustment, the knee was propped with a height-adjustable foam roll.

For all three flexion angles, measurements were performed first without loading and then with a load of approximately 5 kg. The MRI scans were started with a delay of 30 s after the onset of loading to allow adaptation of the cartilage to the load [11,16,22].

SATORI, the web-based application developed by Fraunhofer MEVIS was used for semi-automated segmentation of the MR images. SATORI is a customizable annotation and analysis tool based on the MeVisLab rapid prototyping environment for medical image analysis and visualization [23,24]. Bones and cartilages were semi-automatically segmented in the data acquired at knee extension position (base image). We applied inter-slice smoothing (Gaussian kernel with σ = 1.25 mm) to reduce discontinuities caused by slice-wise drawing.

The Euclidean distance between two opposing cartilage surfaces was computed. Based on these, the cartilage contact area (CCA) was defined as the cartilage surface area with an inter-cartilage distance below 1 mm. Following this definition, we achieved contiguous contact areas without holes in all our measurement data. Accurate and consistent segmentation of the cartilages were needed to compute CCAs for each flexion angle. Based on available manual segmentation masks, a convolutional neural network, known as U-Net, was trained to segment bones in all the remaining images (flexion angle > 0° and loading of 50 N) [25]. An iterative closest point algorithm on the surface meshes created from the masks was used to pre-align the femur (patella) bone in the extended knee to the flexed knee position [26]. Following this coarse alignment of the bones, a rigid image registration of the two MRI images with the normalized gradient fields (NGF) distance measure was performed to perfect the bone alignments [27]. The NGF evaluation was limited to a mask region comprising the femur (patella) bone mask and a dilated (3 mm) region around it, ensuring that the bone contours are entirely accounted for. The resulting refined transformation matrices were used to align femur and patella and the corresponding cartilages in the extension position to the corresponding bone in the flexed knee position.

For statistical analysis, our cohort was split into groups with severe (Dejour type C and D) and mild (Types A and B) dysplastic trochlea. Pre- and post-surgery data were evaluated using a Wilcoxon signed-rank test for paired data. Differences with *p* < 0.05 were considered significant.

Statistical analyses were performed using IBM SPSS Statistics 28.0 (IBM Corp., 2017. IBM *SPSS* Statistics for Windows, Armonk, NY, USA).

This study was approved by the Institutional Review Board (Freiburg University’s Ethics Committee approved this study, ID 443/16) and all subjects provided written informed consent before participation. All subjects voluntarily took part in the study in accordance with the Declaration of Helsinki.

The results of all statistical tests were interpreted in an exploratory sense. No adjustments for multiple testing were made in this exploratory study.

## 3. Results

A total of 13 patients with trochlea dysplasia were included for data analysis. Of those, trochlea dysplasia was classified according to reported ranges in the literature [18,20]. For distributive analysis of mild and severe dysplasia, we conducted a subgroup analysis of patients with mild (Dejour types A and B; *n* = 8) and severe (Dejour types C + D; *n* = 5) trochlear dysplasia. A demographic overview is depicted in Table 1.

The MPFL influence on patellar dynamics was evaluated via the measured differences between pre- and post-surgery patellar tilt, patellofemoral CCA and medial-to-lateral translation of the patella.

### 3.1. Patella Tilt

Data visualized a trending lower patella tilt after MPFL graft installation in both groups and at all flexion angles of the knee (Figure 3 and Figure 4). Results turned out to be significant in every group except for a flexion angle of 15° with loading, as well as 0° extension with loading in case of severe dysplasia. In patients with mild trochlea dysplasia no significant change in patella tilt could be seen in full extension (0°) and 15° flexion of the knee without loading, respectively (Table 2). In patients with mild dysplasia, patella tilt was also significantly lower during 30° flexion with and without loading.

### 3.2. Cartilage Contact Area

The changes in cartilage contact area are depicted in Table 3. We observed a CCA increase in patients with severely and mildly dysplastic trochlea for the loaded and unloaded measurements, compared to pre-surgery measurements (Figure 5 and Figure 6).

However, the increase in CCA at all 0°–15° flexion angles was significant in patients with mild trochlear dysplasia.

In patients with a severely dysplastic trochlea, we perceived a trending increase, but the MPFL graft only had a significant influence on the CCA in 0° flexion (unloaded) and 30° flexion (unloaded and loaded).

## 4. Discussion

To date, this is the first work to have quantitatively assessed patellofemoral dynamics under in situ loading. For the first time, improvement of patellar tilt and CCA could be demonstrated following medial patellofemoral ligament reconstruction for recurrent patellar instability using and A.I. algorithm based on segmented MRI in a loading device. However, in cases of dysplastic trochlea the MPFL did not increase CCA or lowered patella tilt significantly during the flexion of the knee. In an exploratory investigation under in situ loading, Clark et al. measured CCA with patella-femoral instability against healthy individuals with axial MRI sequences across different flexion angles between 0–40° of knee flexion. A continuous increase in CCA was documented over the course of flexion [28]. Similar observations were made in the present study.

Normally, to control the maltracking of the patella before and after surgery, standard imaging assessment comprises plain radiographs, CT scans or MRI scans.

Plain radiographs lack sensitivity in the detection of osteochondral lesions. In approximately 40%, small osteochondral lesions are missed [29]. CT and MRI present modalities to also account for predisposing factors associated with chronic patellar instability or patella maltracking [30,31]. However, all of the above-named imaging procedures only allow static insight into the patella-femoral joint and do not account for dynamic factors, such as quadriceps activation.

The latest literature suggests using a pulley system as a loading device [6,10,14]. Due to limited space in the scanner bore, and to enable load adjustment from the scanner control room, we decided to employ a pneumatic loading device. Motion artefacts were mitigated with camera-based prospective motion correction to optimize the image quality for robust segmentation [32,33].

The main findings of the present research were that the MPFL graft significantly increases the CCA in patients with a mildly dysplastic trochlea for flexion angles of 0°–15° (loaded and unloaded) and significantly decreases patellar tilt with quadriceps activation (loaded) at 15° knee flexion in a mildly and a severely dysplastic trochlea.

In their 2021 study, Stevens et al. reported an increase in the patellofemoral CCA when comparing patients before and after patellofemoral-stabilizing surgery using axial 5 mm slices in loaded dynamic MRI imaging. The greatest differences were shown over a range of 11–20° in active flexion [34]. The results of our study visualize a significant increase in CCA for early flexion angles (0–15°) as well but only in patients with mild dysplasia.

However, the surgical treatment of Stevens et al. comprised trochlea augmentation, tibial tubercule osteotomy and MPFL reconstruction, but in later analysis no distinction between treatment methods was found.

The patella-stabilization process is complex, involving static (joint geometry), passive (supportive retinacula) and active (quadriceps muscles) stabilizers [35]. During the first 20 degrees of knee flexion, the patella is given no bony support, but stabilization is provided by the medial and lateral retinacula—the passive stabilizers [36]. Most of the lateral retinaculum arises from the iliotibial band. Its tightness affects lateral tracking and patella tilt, primarily at around 20° flexion [35,36]. The medial retinaculum is thinner than its lateral counterpart. The MPFL is the primary restrainer to lateral patellar translation. The MPFL contributes up to 60% of the restraint to patellar lateral displacement from full extension to 20° flexion [37]. Beyond 20° flexion, the joint geometry increasingly gains influence on patellar tracking with each degree of flexion [4,38,39]. Whereas the active stabilizer loses influence at higher degrees of flexion, the impact of passive and static stabilizers rises [40,41]. In converse conclusion, the influence of a dysplastic trochlea rises during flexion, causing an alteration in the patellar tilt and reducing the cartilage contact area (CCA), as illustrated by our results. In patients with severe dysplasia, it appears that the degree of dysplasia exceeds the MFPL graft’s capacity to improve the CCA.

### Limitations

This study’s results are limited by the small number of cases. Thus, the results should be understood as exploratory and may vary for larger study populations. As seen and faced by other studies, motion artefacts are major disadvantages of dynamic MRI imaging [22,28,34]. In anticipation we used a prospective motion correction system, which led to a lower rate of “unusable” MRI scans due to motion artefacts.

## 5. Conclusions

This study demonstrates a significant effect of the MPFL graft on cartilage contact area, as well as patella tilt in patients with mild dysplasia of the trochlea. Thus, the MPFL can be used as a single treatment for patients with Dejour type A and B dysplasia.

However, in patients with severe dysplasia the MPFL graft alone does not significantly increase CCA during the flexion phase of the knee. The influence of the trochlea dysplasia does not seem to be compensable with the MPFL alone. Hence, adjunctive measures, such as a surgical trochlea augmentation, seem to be needed with Dejour type C and D.

However, this exploratory study only reports results for very small cohorts. For a better understanding of in situ patella dynamics, more investigations with larger cohorts are needed.

## Figures and Tables

**Figure 1 jpm-12-02049-f001:**
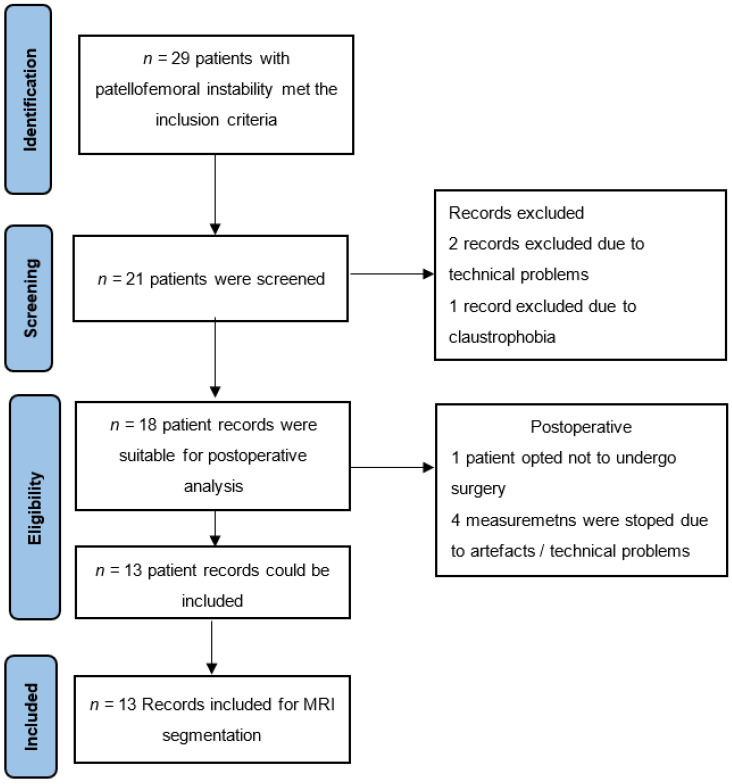
Patient selection process according to STROBE guideline.

**Figure 2 jpm-12-02049-f002:**
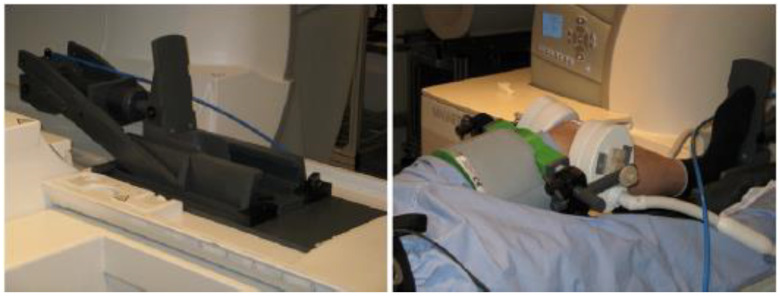
Experimental Setup. The knee is loaded with an MR-compatible pneumatic loading device (**left**). The multipurpose coil is attached to the thigh and the tracking marker for prospective motion correction is taped to the knee cap (**right**).

**Figure 3 jpm-12-02049-f003:**
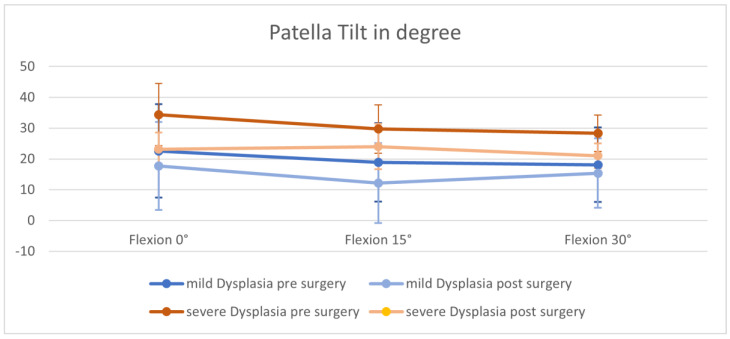
Patella tilt without loading before and after surgery.

**Figure 4 jpm-12-02049-f004:**
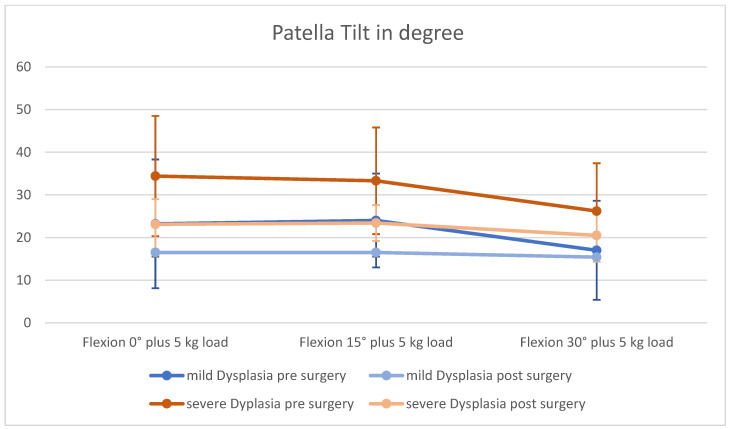
Patella tilt with loading before and after surgery.

**Figure 5 jpm-12-02049-f005:**
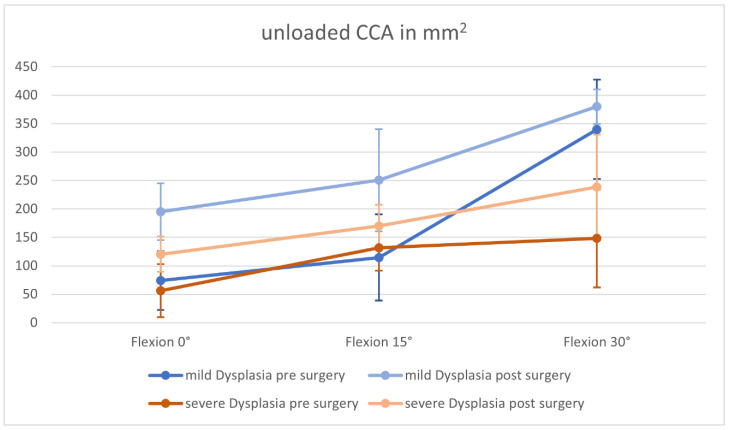
Changes in Cartilage Contact Area (CCA) without loading.

**Figure 6 jpm-12-02049-f006:**
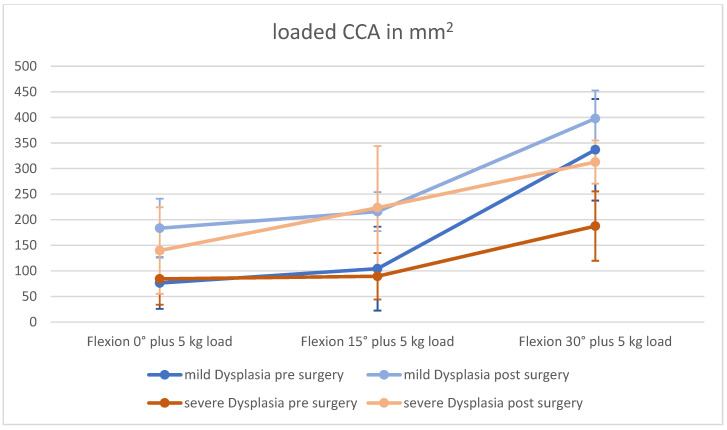
Changes in CCA with loading.

**Table 1 jpm-12-02049-t001:** Cohort demographics.

	Mild Dejour Typ A + B	Severe Dejour Type C + D	Total
Gender ratio Female:Male	4:4	4:1	8:5
Age in years	29.2 ± 9.6	23.5 ± 4.5	27.2 ± 8.4
Height in cm	*171.6* ± *8.2*	*169.8* ± *6.6*	*171.0* ± *7.4*
Weight in kg	*71.5* ± *7.5*	*70.4* ± *5.0*	*71.1* ± *6.6*
BMI	*24.4* ± *2.7*	24.5 ± 2.7	24.2 ± *2.6*

**Table 2 jpm-12-02049-t002:** Patella tilt in degree.

		Mild	Mild			Severe	Severe	
Flexion Angle	n	Pre-Operative Mean ± SD	Post-Operative Mean ± SD	*p*	n	Pre-Operative Mean ± SD	Post-Operative Mean ± SD	*p*
Flexion 0°	8	22.6 ± 15.2	17.7 ± 14.3	0.110	5	34.4 ± 10.1	23.1 ± 7.6	0.043
Flexion 0° plus 5 kg load	8	23.2 ± 15.1	16.5 ± 14.1	0.051	5	34.4 ± 12.1	31.2 ± 16.1	0.500
Flexion 15°	8	18.9 ± 12.7	12.2 ± 13.0	0.052	5	29.7 ± 7.8	24.0 ± 7.4	0.043
Flexion 15° plus 5 kg load	8	24.0 ± 11.0	16.5 ± 12.5	0.044	5	33.3 ± 6.1	23.4 ± 8.6	0.068
Flexion 30°	8	18.1 ± 12.1	15.4 ± 11.3	0.041	5	28.3 ± 5.9	21.0 ± 4.0	0.043
Flexion 30° plus 5 kg load	8	17.0 ± 11.6	15.4 ± 11.2	0.039	5	26.2 ± 6.3	20.5 ± 3.1	0.043

**Table 3 jpm-12-02049-t003:** Cartilage contact area in mm^2^.

		Mild	Mild			Severe	Severe	
Flexion Angle	Number of Patients	Pre-Operative Mean ± SD	Post-Operative Mean ± SD	*p*	Number of Patients	Pre-Operative Mean ± SD	Post-Operative Mean ± SD	*p*
Flexion 0°	8	74.2 ± 51.8	195.1 ± 49.9	*0*.007	5	56.0 ± 46.6	120.5 ± 31.1	0.043
Flexion 0° plus 5 kg load	8	76.3 ± 50.4	183.4 ± 57.7	0.011	5	84.4 ± 50.3	139.8 ± 84.5	0.225
Flexion 15°	8	114.7 ± 75.7	250.4 ± 90.0	0.007	5	131.5 ± 40.3	170.4 ± 36.5	0.345
Flexion 15° plus 5 kg load	8	104.4 ± 82.0	215.9 ± 38.2	0.009	5	89.5 ± 45.4	223.5 ± 120.4	0.068
Flexion 30°	8	339.7 ± 87.3	379.7 ± 30.2	0.128	5	148.1 ± 85.9	238.3 ± 92.5	0.043
Flexion 30° plus 5 kg load	8	336.7 ± 99.3	397.8 ± 54.8	0.056	5	187.6 ± 67.8	312.7 ± 42.2	0.043

## Data Availability

All data are available in the article.

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
