# Peer review of "Individual Influence of Trochlear Dysplasia on Patellofemoral Kinematics after Isolated MPFL Reconstruction"

_jpm, 2022, doi:10.3390/jpm12122049_

Round 1
Reviewer 1 Report
1. In introduction, what does “CCA” stand for?
2. In introduction, please cite reference for why the testing angle between 0, 15, and 30 degrees? Does MPFL play the role during early flexion only?
3. In MM, what is exclusion criteria?
4. In MRI measurements, please draw out the STROBE figure to make the article readable.
5. In line 105, why use “5 kg” for weight bearing? Any reference for citation?
6. In line 183, what does “KI” stand for?
7. In lines 183-184, the summary seems not like your result, as different between “mild and severe” dysplastic trochlea.
8. The content of the discussion is illogical. There is no in-depth discussion of your results. Just stating reference alone.
9. No study limitation; moreover, I DO think the case number is too small to do the statistical analysis.
Overall, the abstract is too long for readability, and the discussion is too superficial.
Author Response
Dear Reviewer,
many thanks for your detailed review. Corrections have been made according to your remarks:
1. In introduction, what does “CCA” stand for?
CCA is the abreviation for Cartilage Contact area, adaption has been made
2. In introduction, please cite reference for why the testing angle between 0, 15, and 30 degrees? Does MPFL play the role during early flexion only?
References have been given and more detailed information added
3. In MM, what is exclusion criteria?
Exclusion criteria have been added
4. In MRI measurements, please draw out the STROBE figure to make the article readable.
STROBE figure has been added
5. In line 105, why use “5 kg” for weight bearing? Any reference for citation?
To simulate wheight beraing and for quadriceps activation without generating unproportional motion artefacts, the load was set to 5kg; Adaptions to the text have been made
6. In line 183, what does “KI” stand for?
translation error K.I = "künstliche Intelligenz" corrected to A.I
7. In lines 183-184, the summary seems not like your result, as different between “mild and severe” dysplastic trochlea.
We compared tables and text and did not find any misleading information. In Case of misunderstanding, could you specify this remark ?
8. The content of the discussion is illogical. There is no in-depth discussion of your results. Just stating reference alone.
Adaption of the Discussion has been made
9. No study limitation; moreover, I DO think the case number is too small to do the statistical analysis.
Limitations have now been stated. We are aware of the small cohort, but the results have to be seen a purely exploratory
We tried to shorten the abstract without losing informtation.
The article was now screened again by a professionell proofreader and antive speaker.
Reviewer 2 Report
Your work is clearly confirming the results for MPFL reconstruction in cases of mild or severe trochlea dysplasia. Patellofemoral dynamics under loading is the most appropriate quantification.
The presence of ligamentous laxity is not mentioned at all in your work. The changes in CCA may be completely different in females with severe laxity.
Measurements in 90d of flexion were not reported at al. Is there any explanation for this. Augmentantion of the dysplastic throchlea may severely alter the dynamics and CCA. This is not commended.
Author Response
Dear Reviewer,
many thanks for your remarks.
Comments to your review are listed below point by point
-The presence of ligamentous laxity is not mentioned at all in your work. The changes in CCA may be completely different in females with severe laxity.
Causes for patellar instability are diverse, but mostly combined if symptomatic. The post-surgery results after MFPL plasty shouldn't be affected by prior laxity, as the MPFL is the major restraining mechanism between angles of 0-30° flexion.
-Measurements in 90d of flexion were not reported at al. Is there any explanation for this. Augmentantion of the dysplastic throchlea may severely alter the dynamics and CCA. This is not commended.
We wanted to proof the capability an MPFL plasty in cases of trochlear displasia, as the MPFL is the major restraint to the patella between flexion angles 0-30°. With flexion more than 30° the ligamentous influence on the patella diminishes. There are also restrictions in the range of motion, which are caused by the physical constraints of an MRI scanner.
Round 2
Reviewer 1 Report
The article is better after revision.